# Efficient Malathion Removal in Constructed Wetlands Coupled to UV/H$_2$O$_2$ Pretreatment

**Cinthia I. G. Cedillo-Herrera** [1], **Adriana Roé-Sosa** [2], **Aurora M. Pat-Espadas** [3], **Karina Ramírez** [1], **Jaime Rochín-Medina** [1] **and Leonel E. Amabilis-Sosa** [4,*]

[1]  Tecnológico Nacional de México/ITCuliacán, División de Estudios de Posgrado e Investigación, Av. Juan de Dios Bátiz 310, Culiacán 80220, Sinaloa, Mexico; cinthia.cedillo@itculiacan.edu.mx (C.I.G.C.-H.); kramirez@itculiacan.edu.mx (K.R.); jaimerochin@itculiacan.edu.mx (J.R.-M.)
[2]  Universidad Tecnológica de Culiacán, Coordinación de Tecnología Ambiental, Culiacán 80014, Sinaloa, Mexico; adriana.roe@utculiacan.edu.mx
[3]  CONACYT-UNAM Instituto de Geología, Estación Regional del Noroeste, Avenida Luis D. Colosio esquina Madrid, Hermosillo 83000, Sonora, Mexico; apat@conacyt.mx
[4]  CONACYT-Tecnológico Nacional de México/ITCuliacán, División de Estudios de Posgrado e Investigación, Av. Juan de Dios Bátiz 310, Culiacán 80220, Sinaloa, Mexico
*  Correspondence: lamabilis@conacyt.mx

**Abstract:** Intensive agriculture has led to the increasing application of pesticides, such as malathion, thus generating large volumes of untreated cropland wastewater (CropWW). In this work, a hybrid system constructed wetlands (CW) coupled in continuous with an optimized UV/H$_2$O$_2$ pretreatment was evaluated for the efficient removal of malathion contained in CropWW. In the first stage, 90 min UV irradiation time (UV IR) and 65 mM hydrogen peroxide (H$_2$O$_2$) were identified as optimal operation parameters through a central composite design. The second stage consisted of CW planted with *Phragmites australis* collected from the agricultural discharge area and operated as a piston flow reactor. Furthermore, CW hydraulic residence times (HRT) of 1, 2 and 3 days, including hydraulic coupling, were evaluated. The removal efficiencies obtained in the first stage (UV/H$_2$O$_2$) were 94 ± 2.5% of malathion and 45 ± 2.5% of total organic carbon (TOC). In stage two (CW) 65 ± 9.6% TOC removal was achieved during the first 17 days, from which around 24% was associated to the biosorption of malathion byproducts. Subsequently, and until the operation ends, CW removed about 80% of TOC for 2 and 3 days HRT, with no significant differences ($p > 0.2$), which is higher than those reported in several studies involving only advanced oxidation processes (AOP) with UV IR times above 240 min and even for systems using catalysts. The results obtained indicate that the system UV/H$_2$O$_2$-CW is a technically suitable option for the treatment of CropWW with a high content of malathion mainly found in developing countries. Moreover, the hybrid system proposed also represent significant reduction in the size of the treatment plant.

**Keywords:** constructed wetlands; cropland wastewater; malathion; AOP optimization

## 1. Introduction

Extensive agricultural production implies continuous use of agrochemicals to guarantee pest control and maximize crop yields. Since the 1970s, pesticides have been detected in environmental matrices such as soil, water and in organisms at practically all levels of the food chain due to their bioaccumulative and biomagnifiable characteristics. Even more relevant are their carcinogenic, mutagenic and teratogenic effects [1,2]. Pesticides are classified based on their chemical structure and functional groups as carbamates, organochlorines, organophosphates, pyrethroids and bipyridyls.

Currently, organophosphate pesticides are the most widely used insecticides in tropical regions, representing health risks since their mechanism of action inhibits the cholinesterase enzyme [3].

Around seventy percent of the extracted water volume is used for agriculture, hence this economic activity is considered the largest water consumer. In developing countries this volume reaches up to 95%, because irrigation practices do not allow optimal water use [4]. Consequently, a considerable volume of cropland wastewater (CropWW) is generated, which is characterized by low biodegradability and a high concentration of nutrients [5,6]. The CropWW is transported through the infrastructure of agricultural drains to surface water bodies, sometimes ending in the aquifer that supplies water for the population. In both cases, the composition and volume of CropWW discharged generates diffuse discharges difficult to control which causes pollution problems, eutrophication and health risks.

Cereals are one of the primary sources of food around the world. In 2018/19 agricultural cycle, their production amounted to 2653 million tons with an estimated increase of 2.3% for the following cycle [7]. In tropical regions, maize requires between 500 and 800 mm of water layer per cycle [8]. During the cultivation stage, malathion is intensively applied for insect control which is subsequently released into the environment through agricultural drains. Despite its half-life of less than one week, this pesticide has been quantified in agricultural drains and in the vicinity of water bodies and river mouths of neighboring water bodies in concentrations ranging from 0.005 to 3.3 µg/L [9,10].

Chemical formula of malathion is $C_{10}H_{19}O_6PS_2$, and it presents chronic and environmental toxicity characteristics; its $LC_{50}$ for rats by inhalation is 43.8 mg/m$^3$/4 h (PubChem, 2019) [3]. This pesticide class is not as persistent as organochlorines, but its degradation byproducts, such as malaoxon, have a higher bioactivity and toxicity [11,12].

Moreover, CropWWs have a high load of nitrogen and phosphorus from fertilizers, thus causing eutrophication problems in agricultural basins [13]. Therefore, implementation of effective and feasible CropWW treatment is crucial to mitigate water bodies' pollution.

Ref. [14] reported removal efficiencies of chlorpyrifos, lambda-cyhalothrin and diazinon of 73.3%, 68.9%, 49.1% and 84%, respectively, as chemical oxygen demand (COD) through a photocatalytic process of wastewater from an agrochemical and pesticides company. The authors reported 120 min as optimal irradiation time, and $H_2O_2$ and $TiO_2$ concentrations of 1.5 g/L and 2 g/L, respectively.

Ref. [15] measured atrazine degradation (compound with toxicity and application volume similar to malathion) by a UV/ $H_2O_2$ system, reaching a complete removal after 30 min of irradiation with 20 mg/L $H_2O_2$. However, the concentration of non-biodegradable byproducts increased over time.

Ref. [16] tested organophosphate pesticide removal by different advanced oxidation processes (AOP), such as UV/$H_2O_2$, photo-Fenton and ozonation, using concentrations up to 12 and 11 mg/L of malathion and parathion. All treatments tested removed around 100% of both pesticides at 90 min of operation. However, AOP needed high irradiation times (120–300 min) to achieve mineralization, resulting in high operating costs and reagent usage.

On the other hand, constructed wetlands (CWs) are known for their efficiency and versatility in wastewater treatment by the interaction of physical, chemical and biological processes, such as oxidation, reduction, precipitation, sedimentation and adsorption [17]. Recently, these biological treatments have been used for industrial effluents, mine and agricultural drainages as well as the removal of other persistent organic pollutants [18]. Likewise, natural wetlands around cropland drainages have demonstrated to perform a feasible treatment [19].

Because CropWW is characterized by its low biodegradability, conventional CWs are technically limited. Nonetheless, this technology has evolved for treating emerging contaminants by using emerging vegetation and/or local bacterial communities improving their efficiency as bioaugmented CWs [20]. However, this type of CW has not been scaled up mainly because they need high HTR, which implies large land areas.

The technologies available for the treatment of wastewater containing pesticides, for instance AOP and biological systems, offer advantages and some degree of efficiency. Hence, the proper coupling of

both technologies would suggest an increase in the removal efficiency and the possible reduction of toxic byproduct formation, as well as reduced area requirements for CWs.

Ref. [6] studied the coupling of CW to photo-Fenton, ferrioxalate and photo-Fenton/$TiO_2$ for the removal of clopyralid contained in synthetic wastewater. Total organic carbon (TOC) removal reached 93, 92 and 87% respectively, almost twice as high as the treatment before the coupling with the biological process. Similarly, [21] coupled a photocatalytic system $TiO_2$-visible light with CW to treat decabromodiphenyl ether. Removal achieved in the combined system (93.6 ± 2.2%) was significantly higher than in the photocatalytic system alone (56.3 ± 2.8%).

The aforementioned, suggest the technical feasibility of treating CropWW using AOP coupled to CWs, since biological systems are characterized by operational robustness to low pH values and byproducts likely to be present in the effluent of AOP. In this context, it is possible to achieve high removal rates, or even mineralization, of organic matter producing an effluent with acceptable characteristics for reuse in the same agricultural areas.

In this study the main objective was to assess technical feasibility of a $UV/H_2O_2$-CW coupled system to treat CropWW with high malathion and nutrient loads, which is discharged into agricultural drainages in tropical regions [13]. Before coupling, the AOP was optimized to achieve the highest removal rates as well as a lower energy consumption.

## 2. Materials and Methods

The hybrid treatment system set up consisted of an $UV/H_2O_2$ process coupled to CW. The $H_2O_2$ dose and irradiation time were optimized for the AOP. The coupled system was operated in continuous flow, thus the AOP's effluent was the influent of the CW. Figure 1 shows the general diagram of the coupled treatment system, Sections 2.1–2.3 give further details of each process including hydraulic coupling.

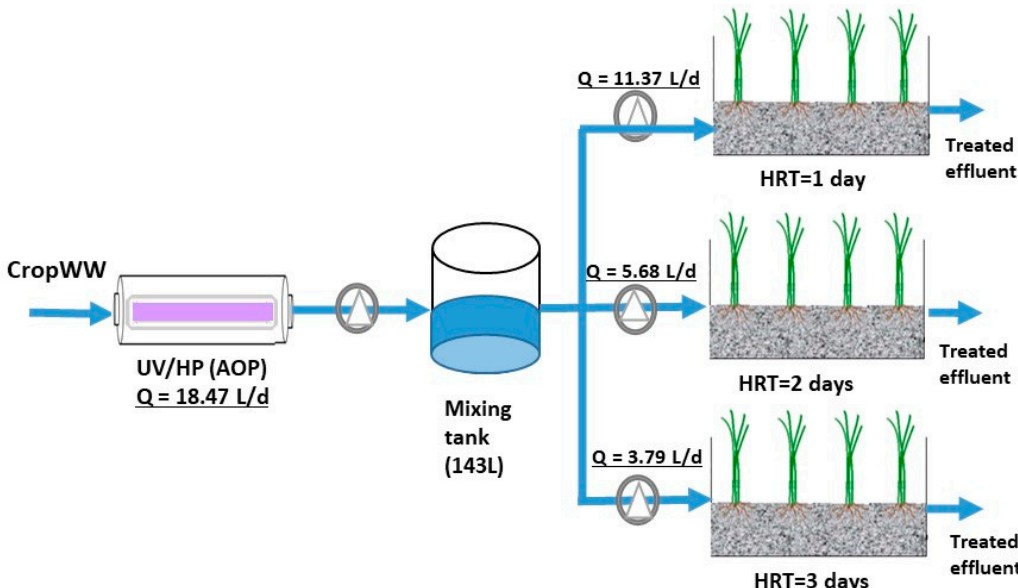

**Figure 1.** Hybrid system general diagram during experimental setup and operation. Q = volumetric flow and HRT = hydraulic residence time.

The influent of hybrid system was synthetic water with the following composition: 41.8 mg/L malathion (commercial malathion), equivalent to 22 mg/L of TOC [16], 2.12 mg/L total phosphorus (TP) (added as $KH_2PO_4$) and 0.17 mg/L of total nitrogen (TN) (added as $KNO_3$). The concentrations used were established according to the the maximum effluent concentrations found in crop areas where malathion and fertilizers are widely used [22].

### 2.1. Pretreatment UV/$H_2O_2$ Optimization

The AOP UV/ $H_2O_2$ pretreatment was carried out in a stainless-steel recirculation chamber with a diameter of 7 cm and 30 cm length. A 10 W UV mercury lamp at 254 nm coated with optical grade quartz, was installed inside the chamber (Figure 2). Into the UV chamber, $H_2O_2$ (30% *w/w*, industrial grade) was dosed using a peristaltic pump, Kaomer 5V. In order to optimize the system, 10, 30, 60, 90, 120 and 150 min of UV irradiation and $H_2O_2$ dosages of 49, 65 and 82 mM were evaluated, considering previous studies on recalcitrant compounds removal by AOPs, in which the kinetics of hydroxyl radicals (•OH) generation were studied [16,23]. The relationship between irradiation time and $H_2O_2$ dose was analyzed through the Response Surface methodology with a central composite design using malathion and TOC concentrations as a response variable and desirability function using all quadratic effects interactions.

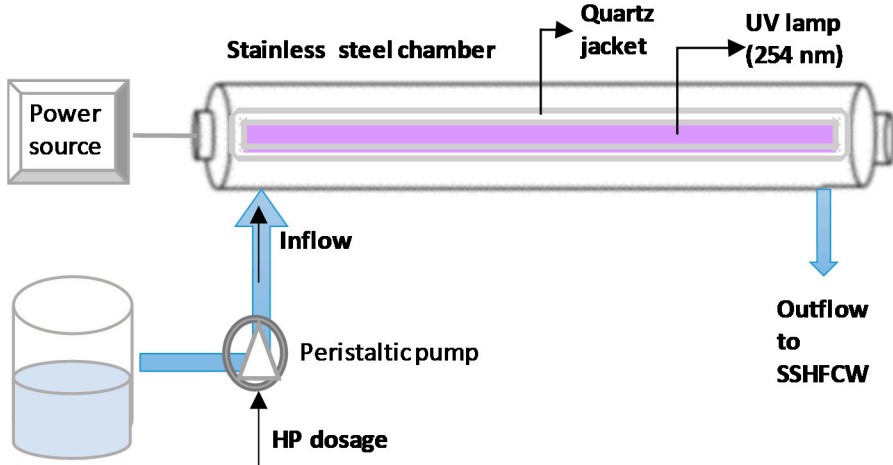

**Figure 2.** Advanced oxidation process UV/$H_2O_2$ used as constructed wetland system pretreatment. HP = peroxide hydrogen and SSHFCW = subsurface horizontal flow constructed wetland.

### 2.2. Design and Operation of CW

The biological treatment was performed at laboratory scale in subsurface horizontal flow constructed wetlands (SSHFCW). Bioreactors were constructed of acrylic with 0.55 m length, 0.20 m width and 0.35 m total height (Figure 3), with a length-to-width ratio of 2.7:1, proportion suggested in several studies to set up piston flow [24].

The support medium consisted of gravel with a particle diameter of 1.0 cm, equal to 43.33% of porosity [25]. Therefore, the effective volume was 11.37 L. The water inlet was positioned 5 cm below the top surface layer of the support medium and distilled water was added to maintain the operating volume to avoid the increase of dissolved organic matter concentration due to water losses by evapotranspiration.

CWs were planted with *Phragmites australis*, an adaptable specie to climatic and operational conditions [26]. Morevover, this is the most common vegetation growing in the vicinity of cropland drainages. One stem was planted considering the superficial area as per the recommendations by [20] (see Table 1).

**Table 1.** Constructed wetland (CW) operational conditions. In vegetation density, 9 stem/m$^2$ is resulted of planting one stem in the 0.11 m$^2$ of superficial area.

| Features | Units | Bioreactor 1 | Bioreactor 2 | Bioreactor 3 |
|---|---|---|---|---|
| Cross section area | m$^2$ | 0.07 | 0.07 | 0.07 |
| Vegetation density | Stem/m$^2$ | 9 | 9 | 9 |
| Hydraulic residence time | days | 1 | 2 | 3 |
| Organic load | gCOD/(m$^2$ d) | 12.22 | 6.11 | 4.07 |

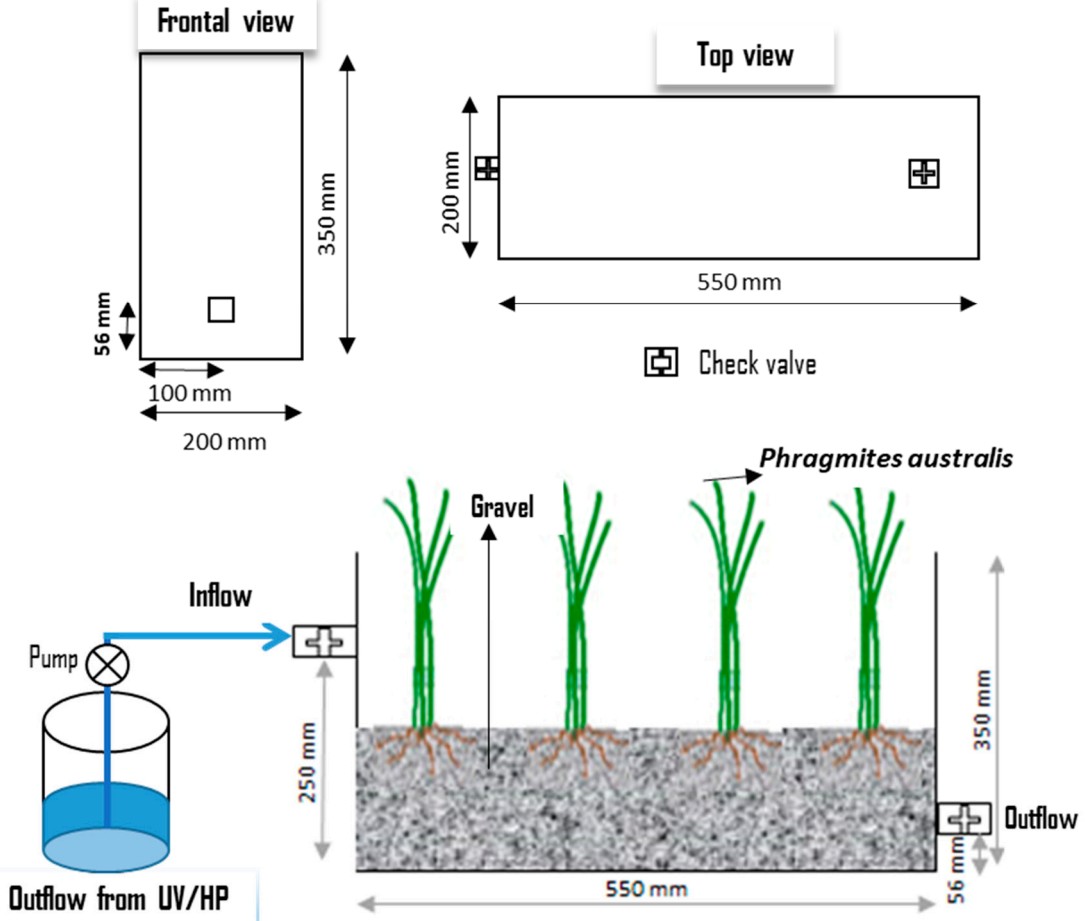

**Figure 3.** Schematic diagram of experimental subsurface horizontal flow constructed wetlands.

Previous to the coupling of the processes, the biological system was adapted in two phases. The first phase involved hydroponic cultivation of transplanted vegetation from an agricultural drainage. In the second stage vegetation was planted in CW and fed with municipal wastewater collected from the primary sedimentation tanks of a local treatment plant. The influent to the CW was changed to CropWW when the volatile suspended solids concentration was steady [27]. Table 1 shows the main design and operational parameters of the CW system.

### 2.3. UV/$H_2O_2$-CW System Coupling and Analytical Procedures

The flow rate of pre-treatment UV/$H_2O_2$ was 18.47 L/d, considering the optimum irradiation time and the photo-reactor volume (1.154 L). Afterward, the UV/$H_2O_2$ effluent was conducted to a 143 L homogenization tank, connected to CW (See Figure 1). The effective volume of CW was 11.37 L and hydraulic residence times (HRT) of 1, 2 and 3 days were evaluated. Hence, the removal was calculated according to the total UV/$H_2O_2$-CW coupled systems. Every system evaluated was integrated with the same AOP pretreatment, but the CW was operated at different HRT values.

Synthetic CropWW was conducted through the system by a digital filling peristaltic pump CR BT100FJ with silicone tubing with inner diameter 0.76 mm. The hybrid system was operated for 60 days to achieve stable organic matter removal efficiencies, which must be observed in CW operated under continuous flow [27,28].

Malathion concentration was measured with an Agilent 7890A gas chromatography equipped with an HP-5 capillary gas chromatographic column and a Flame Photometric Detector. Nitrogen gas used as a carrier and operation parameters such as a temperature of 300 °C during the chromatographic cycle, a purge time of 0.75 min and injection volumes of 4 μL were according to [29].

Total organic carbon (TOC) was quantified according to American Public Health Association [30] with a LECO CR-412 analyzer TOC-L equipment. The pH and oxidation–reduction potential (ORP) in the water samples were measured using a multi-parameter tester (Hanna HI9828). These water quality parameters were monitored during the different stages of the hybrid system.

### 2.4. Statistical Analysis

The AOP optimization tests were run with five replicates, and the coupled system $UV/H_2O_2$-CW was performed in triplicate. The data obtained from the coupled systems were compared using repeated measures analysis of variance (ANOVA) followed by a Tukey test ($\alpha = 0.05$).

Response variables were TOC and malathion concentration. Previously, data were evaluated for normality and homoscedasticity with the Anderson–Darling test. The entire statistical analysis was performed using Stat Graphic Centurion XVI.

## 3. Results and Discussion

### 3.1. Advanced Oxidation Process $UV/H_2O_2$

#### 3.1.1. Malathion and TOC Removal

Figures 4 and 5 show the behavior of malathion and TOC concentration in the AOP. For the three $H_2O_2$ doses, malathion removal reaches its highest level, $95 \pm 3.2\%$, with an asymptotic tendency after 90 min of irradiation. However, TOC removal around 50%–60% at 90 and 120 min of irradiation, with $H_2O_2$ doses of 82 and 65 mM, respectively, were obtained. The tendency of high malathion removal but low TOC removal (22%–40%) was also observed for the 49 mM $H_2O_2$.

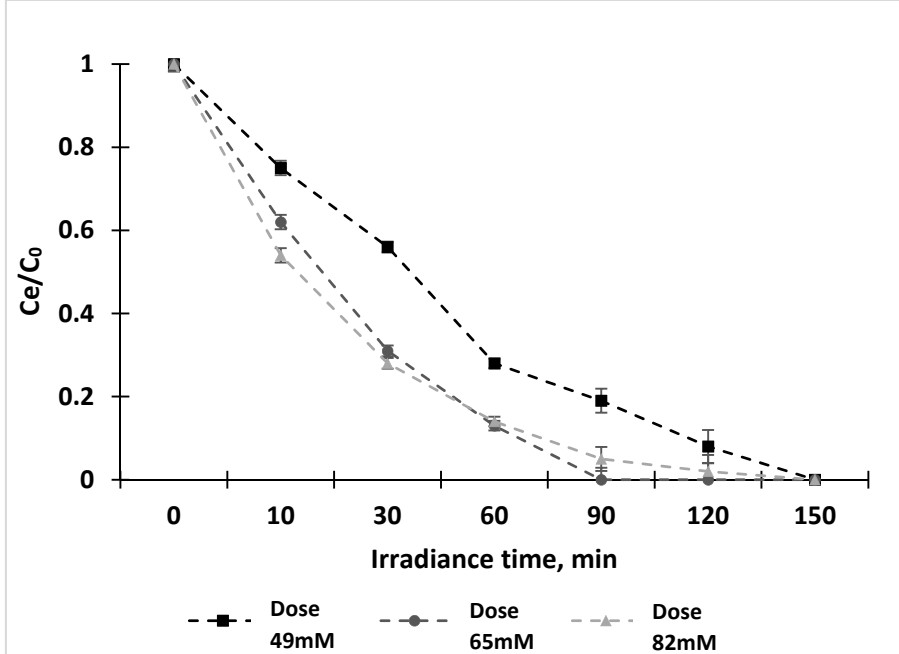

**Figure 4.** Removal of malathion by $UV/H_2O_2$-advanced oxidation process at three different $H_2O_2$ dosage. The values represent the mean values of five replicates with their standard deviation. Ce = malathion concentration at time t (0, 10, 30, 60, 90, 120 or 150 min), $C_0$ = initial malathion concentration (In this research = 22 mg/L).

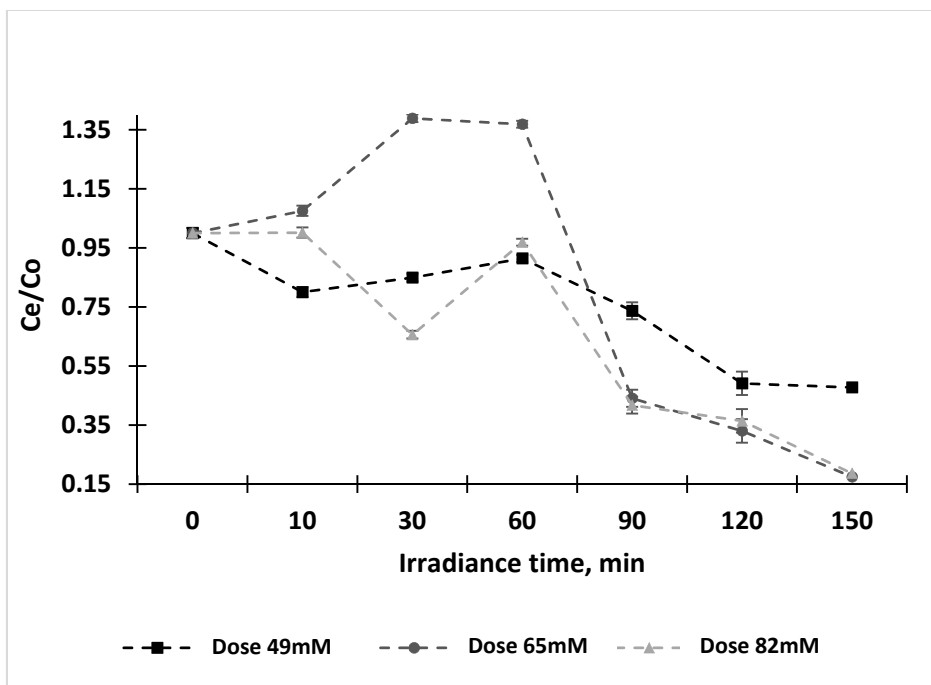

**Figure 5.** Total organic carbon (TOC) behavior during malathion removal by UV/$H_2O_2$-advanced oxidation process at three different $H_2O_2$ dosage. The values represent the mean values of five replicates with their standard deviation. Ce = TOC concentration at time t (0, 10, 30, 60, 90, 120 or 150 min), $C_0$ = initial TOC concentration (In this research = 41.8 mg/L).

The differences between malathion and TOC removal behavior is related to the degradation of pesticide to less complex organic molecules, in terms of chemical structure, but together contributes to the same organic carbon concentration [31]. Regarding TOC removal, it is only observed if the degradation product is $CO_2$, equivalent to mineralization. Results showed that the malathion molecule was practically not detected (Figure 4) at 90 min for 65 and 82 mM $H_2O_2$ doses, but the degradation byproducts were quantified as 40 ± 3.2% of TOC initial concentration (Figure 5). Organic byproducts of malathion degradation by AOP are maloxon, malathion dicarboxylate, dimethyl dithiophosphate, dimethyl thiophosphate and methyl thiophosphate, which are quantified as TOC [32].

The formation of the aforementioned byproducts depends on the hydroxyl radicals (•OH), which in photolysis are correlated to the initial concentration of the oxidant reagent, in this case $H_2O_2$ [33]. According to the results shown in Figures 4 and 5 the removal tendency for malathion and TOC is remarkably similar for 65 and 82 mM $H_2O_2$ ($p = 0.09$), particularly, after 90 min of irradiation. The results for both doses of $H_2O_2$ did not show significant differences ($p = 0.13$).

Results from the lower dose tested of 49 mM $H_2O_2$ demonstrated that 100% of malathion removal can be achieved but it requires 150 min of irradiation (Figure 4). However, this condition exhibited the highest TOC removal at 120 min ($p = 0.009$), which is associated to $H_2O_2$ depletion, since the production of •OH is limited when consumed completely.

The UV/ $H_2O_2$ system achieves its highest removal efficiency at 130 min of irradiation, suggesting consumption of most of the reagent; therefore, a longer HRT does not increase its removal rate (Figures 4 and 5).

The optimal $H_2O_2$ dose is in a range 65–82 mM. Thus, addition of $H_2O_2$ generates •OH that enhances the efficiency of dissolved organic matter oxidation [34,35]. Nevertheless, increase reagent amount may affect degradation, since excess peroxide molecules cause competing reactions with organic matter by •OH, and thus their depletion [36].

### 3.1.2. Behavior of Physicochemical Parameters

The pH levels in the system remained in acidic condition, below 2.5 units during the AOP (Figure 6a). Such conditions enhance organophosphate pesticides degradation through advanced oxidation processes [37]. The $H_2O_2$ oxidation generates •OH, and in turn releases hydronium ions, responsible for the medium acidification. According to the study results, pH and TOC removal decrease over time (Figures 4, 5 and 6a).

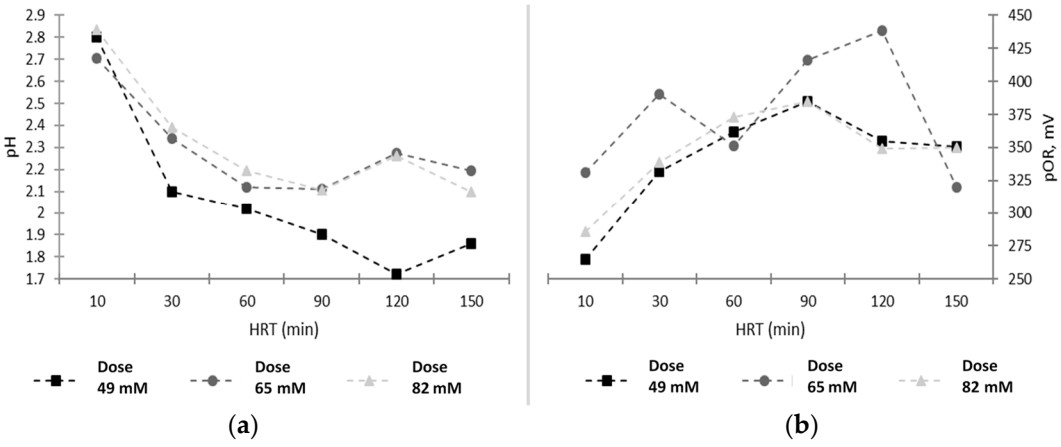

**Figure 6.** Pshysicochemical parameters behavior in UV/$H_2O_2$-advanced oxidation process at three different $H_2O_2$ dosage during malathion removal. The values represent the mean values of five replicates. (**a**) pH and (**b**) oxidation–reduction potential.

Figure 6b indicates that oxidative conditions were maintained throughout the treatment. Although ORP values did not show a clear trend linked to irradiation time, higher ORP values were observed for the 65 mM of $H_2O_2$, which is mainly because $H_2O_2$ determines oxidative conditions of the treatment regardless organic matter degradation and/or oxidation of dissolved solids [38,39]. Indeed, efficiencies of malathion and TOC removal with 65 mM were higher than with 82 mM $H_2O_2$, despite no statistical difference ($p = 0.11$). Ref. [40] have evaluated the doses of peroxide in advanced oxidation processes, finding that when the kinetics of decomposition, manifested as a greater amount of •OH, are greater than degradation kinetics, there is competition for UV light.

### 3.1.3. Selection of Pretreatment UV/$H_2O_2$ Operation Parameters

Figure 7, based on the results of Section 3.1, shows the interaction effects of irradiation time and $H_2O_2$ dose on the organic matter removal for optimization. In terms of energy and oxidant reagent consumption, the combination that allows a shorter irradiation time and lower consumption of reagent without compromising the pretreatment removal efficiency was considered.

Hence, 75 and 120 min of UV irradiation and $H_2O_2$ dosage between 49 and 65 mM were the optimal conditions to achieve satisfactory removal efficiencies for the subsequent treatment stage. Table 2 describes the more favorable treatment configurations, which were selected to be 90 min with 65 mM, since same removal efficiencies are achieved with higher irradiation time and/or concentration of $H_2O_2$. The second option, 90 min and 57 mM, was excluded because TOC and malathion removal efficiencies are lower in comparison with the third option ($p = 0.008$). Furthermore, such efficiencies are practically equal to the first option, which use shorter irradiation time and reagent volume (Table 2). The third option was selected over the first one, despite the longer irradiation time, because removal efficiency was 50% higher for both malathion and TOC. In this regard, the lowest concentrations of TOC and malathion to be treated in the biological systems were considered in order to reduce the organic load to the CWs which have better performance at low organic loads [5,39].

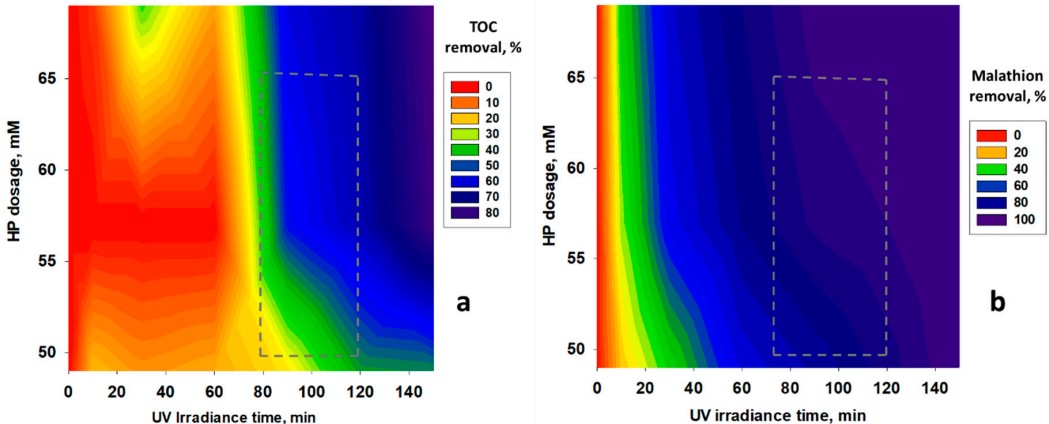

**Figure 7.** Effect of the combination of $H_2O_2$ doses and different UV irradiation times over the concentration of (**a**) TOC and (**b**) malathion. The dotted line box indicates the region with high removal but takes into consideration the shortest possible irradiation times and peroxides dosage.

**Table 2.** Potential combinations of irradiation time and $H_2O_2$ dose during the pretreatment operation, considering both malathion and TOC removal. Note: This table is modified from the result of the response surface design analysis, where the confidence intervals are bilateral to estimate the effect of higher and lower levels.

| Pretreatment Option | Irradiation Time, Min | Hydrogen Peroxide, mM | TOC Removal, % | Malathion Removal, % |
|---|---|---|---|---|
| 1 | 75 | 49 | 30–40 | 70–80 |
| 2 | 90 | 57 | 40–50 | 70–80 |
| 3 | 90 | 65 | 50–60 | 90–100 |
| 4 | 120 | 65 | 50–60 | 90–100 |

### 3.2. UV/$H_2O_2$-SSHFCW System

The synthetic CropWW initial concentration ($C_0$), for configurations of the hybrid system, was 41.8 mg/L of malathion and 22 mg/L of TOC. In this section, the only difference between the three UV/$H_2O_2$-CW systems evaluated was the HRT in the CW, as it was described in Section 2.3. Figure 8 shows the removal rates in the UV/$H_2O_2$-CW systems.

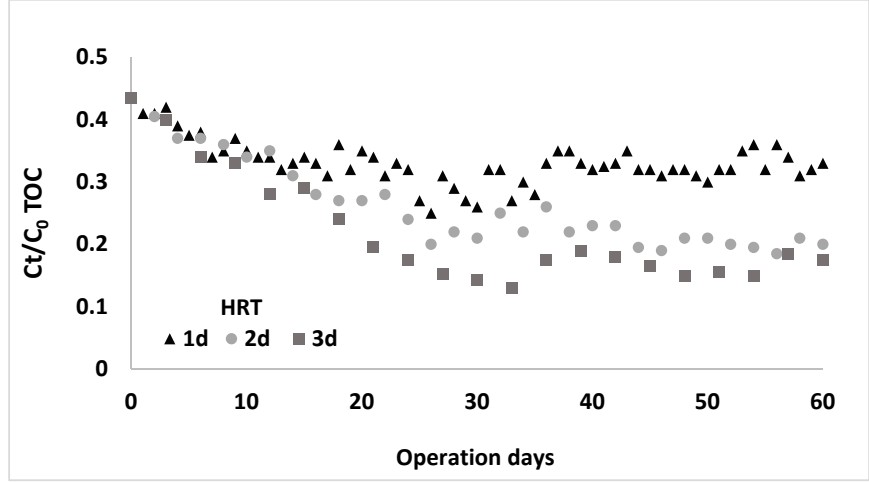

**Figure 8.** TOC behavior dosage during malathion removal in hybrid UV/$H_2O_2$-CW system at three hydraulic retention times used for the CW stage. The values represent the mean values of three replicates. Ce = TOC concentration at time t (operation days), $C_0$ = initial TOC concentration (the value corresponds to the AOP effluent).

Malathion removal in the hybrid system was 94.7 ± 4, 97.9 ± 2 and 98.3 ± 3% for the wetland HRTs of 1, 2 and 3 days, respectively. Although higher pesticide removal was observed at longer HRT in CW, there was not significant difference ($p = 1.2$), since removal mainly occurred in the AOP pretreatment which was the same for all three systems. In fact, this pretreatment achieved between 95% and 100% of malathion removal with the operational parameters selected.

Regarding TOC, the three CW configurations coupled to UV/$H_2O_2$ pretreatment achieved 65 ± 8.3% TOC removal at 17 operation days, which is 15% more than the AOP as the unique treatment. The results obtained suggest that a portion of malathion by-products, generated during the AOP, were removed in CW by some mechanism other than biodegradation, since HRT would influence the removal efficiency in biological systems (time contact microorganisms-contaminant). Some authors have reported that particles or surfaces of organic–inorganic nature can adsorb organic persistent molecules such as pesticides, drugs and their cyclic byproducts by Van Der Waals forces and/or physisorption, as occurs in soils and sediments [42,43]. In CWs, sorption processes can occur in support media, due to occluded and interstitial porosity. In addition, sorption also occurs on hydroxyalkanoates and hydroxybutyrates, main compounds of microbial biofilm [44,45] and onto the root exudates of vegetation rhizomes [44,46].

Regarding TOC removal, the performance of the UV/$H_2O_2$-CW operated with 1 d HRT was significantly lower than the other two systems ($p = 0.0015$), which is mainly because the recommended HRT for CW range from 1.5 to 4 days.

In fact, in a non-aerated attached-biomass reactor, catabolic processes are not observed before one day HRT [18,47]. Furthermore, this hybrid system did not show any trend throughout the experiment (Figure 8), so it is likely that the removal achieved in the CW was due to sorption and biosorption.

The UV/$H_2O_2$-CW operated with 2 and 3 d HRT have no significant differences between removals ($p = 0.2$). Both systems were stable before 40 days of operation when the maximum variation is ±10%, acceptable for secondary wastewater treatment systems [18]. TOC removal efficiency achieved in both systems was 80 ± 6.9%, which is higher than the value reported for biological systems treating pesticides [48] and similar to that reported for AOP operated with high irradiation times of 150–300 min [16–49].

The similar results obtained for systems operated with 2 and 3d HRT during the second stage, was probably due to the recalcitrant nature of some pesticide byproducts. Therefore, microorganisms can no longer degrade or mineralize the compounds even after a long time contact, thus TOC concentration does not decrease with higher HRT. Some studies have suggested the degradation of malathion to phosphorylated thiol groups before mineralization [11,12], but these compounds are barely biodegradable due to their bactericidal effect [50].

On the other hand, authors have highlighted that the importance of vegetation in CW resides in the rhizosphere, especially in rhizospheric bacterial strains [51,52]. The vegetation planted in the CWs was collected from agricultural drains where organophosphate pesticides and traces of organochlorines have been reported [22]. This suggests that rhizospheric bacteria could be tolerant to pesticides and even assimilate them, as it has been demonstrated for in vitro studies [53] and bioreactors [54] that the adapted bacteria promoted higher removal rates and, therefore, lower HRT requirements. Nonetheless, this fact requires evaluation in future research.

## 4. Conclusions

A sequentially coupled UV/$H_2O_2$-CW was successfully used for the treatment of synthetic cropland wastewater containing malathion. Based on the results, the following points are concluded:

- In the first stage, the UV/$H_2O_2$ system optimization resulted in the removal of most malathion content in a shorter time of period than the reported for other AOP, including those using catalysts. The TOC remaining after this stage was mainly attributed to the formation of byproducts.
- The UV/$H_2O_2$-CW system removed an average of 65% TOC during the first 17 operation days without significant difference ($p > 0.05$) among the HRTs used in constructed wetlands related to

biosorption processes. Nevertheless, the system with the lower HRT value did not increase its efficiency or reach stability over time.

- The UV/H$_2$O$_2$-CW system operated with 2 and 3 d HRT in CW, reached 80 ± 6.9% TOC removal, which is equivalent to value achieved by novelty prototypes based on the combination of two or more AOP.
- The overall results demonstrated that CWs planted with native vegetation from cropland and coupled in continuous flow to UV/H$_2$O$_2$ pretreatment is a technically feasible option to treat CropWW containing malathion.
- Future research efforts regarding toxicity, degradation pathways and quantification of other recalcitrant compounds, even at trace levels, are needed to better understand the treatment process.
- Long term evaluation of in situ pilot-scale systems treating real wastewater would be necessary to evaluate the economic feasibility of the large-scale system. An economic comparison between the proposed hybrid system and the conventional advanced oxidation will help to confirm if the one proposed in this research is less costly as is expected.

**Author Contributions:** C.I.G.C.-H.: Data curation and writing original draft. A.R.-S.: Data curation and conceptualization. A.M.P.-E.: Writing, review and editing. K.R.: Writing, review and editing. J.R.-M.: Data curation and methodology. L.E.A.-S.: Conceptualization, methodology, writing-original draft and project administration. All authors have read and agreed to the published version of the manuscript.

**Funding:** This research was financially supported by CONACYT projects Problemas Nacionales 2017-1 ID 5020 and is part of Cátedras CONACYT Ref. 2572.

**Acknowledgments:** The first and second authors acknowledge the technical facilities provided through to Proyecto interno de la Universidad Tecnológica de Culiacán 2019–2020. Likewise, all authors are grateful for the support given by Asociación de Usuarios Productores Agrícolas El Grande Módulo II-3.

**Conflicts of Interest:** The authors declare no conflict of interest.

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
