# Peer review of "Efficient Malathion Removal in Constructed Wetlands Coupled to UV/H2O2 Pretreatment"

_applsci, doi:10.3390/app10155306_

Round 1
Reviewer 1 Report
The manuscript of Cedillo-Herrera et al. investigates efficiency of a pesticide removal by hybrid constructed wetlands. The manuscript is clearly and well written. I only recommend English proofreading to the authors, especially for the abstract.
Author Response
Reviewer 1
Comments and Suggestions for Authors
The manuscript of Cedillo-Herrera et al. investigates efficiency of a pesticide removal by hybrid constructed wetlands. The manuscript is clearly and well written. I only recommend English proofreading to the authors, especially for the abstract.
Dear Reviewer 1,
R = The authors are grateful for your comments and recommendations. In this regard, we have carefully reviewed the English, especially the proofreading of the abstract.

Reviewer 2 Report
Authors present an interesting work that can be useful for the treatment of wastewater contaminated with pesticides.
Some minor revisions are needed before being accepted
Abstract:
line 27: 80% of TOC?
line 30: capacity of what?
Introduction
lines 37,38: not clear, authors mention the pesticides were detected in water and soil. how is that related with food chain? Please clarify and do the necessary amendments
line 106: If this is known, this needs a reference.
Material and Methods
Figure 1: defined in the figure caption what is Q, and what is HRT
lines 130, 131: why were these minutes of UV radiation chosen? Why these H2O2 dosages? Please, clarify and add relevant information
Figure 2: indicate in figure caption what is HP and SSHFCW
Table 1: defined what is HRT
Line 169: indicated that it is synthetic cropWW
Line 170: why 60 days?
Line 171: how was the methodology for malathion determination optimised. Optimisation parameters should be included or a reference should be added
Line 175: defined ORP
Line 176: at what type samples were collected? only after the 60 days? the effluent of CW was discarded? There was always input of new AOP effluent to CWs?
Results
Figurw4: this caption need more information. What is being shown in yy axis? what is Ce/Co? Removal in what units? Values presented are mean values with their standard deviation? how many replicates? all these information need to be in figure legend
Figure5 : this legend need more information, please, see comments for figure 4 and make the necessary amendments
Fiigure6: define pOR. Are these mean values?~
Table 2: how was this surface design analysis done? Please include relevant information
Line 263: indicate that it is synthetic CropWW
Line 275: “…by some mechanisms other than biodegradation” - this is not clear. Why not biodegradation? Please, clarify and add relevant information
Figure 8: please, see comment for previous figures legend and include more information accordingly
Conclusions
Line314: indicate that it is synthetic croplant wastewater
Line 332: not only this but test should be done also with real wastewater. Please, include. Besides, some discussion should be done regarding the economical aspect in the sense that an hybrid system can be less costly than single AOP
Author Response
Reviewer 2
Comments and Suggestions for Authors
Authors present an interesting work that can be useful for the treatment of wastewater contaminated with pesticides.
Some minor revisions are needed before being accepted
Abstract:
line 27: 80% of TOC?
R= Thanks for the observation. Is correct, 80% of TOC. The omission has already been corrected in the sentence.
line 30: capacity of what?
R= Thanks for the observation. Treatment capacity. The word has already been included in the sentence.
Introduction
lines 37,38: not clear, authors mention the pesticides were detected in water and soil. how is that related with food chain? Please clarify and do the necessary amendments
R= Thanks for the observation. The wording was improved to clarify that pesticides have been quantified in environmental matrices such as soil, water and organisms. The text specifies that organisms from practically all levels of the food chain.
line 106: If this is known, this needs a reference.
R= Thanks for the observation. The authors agree. The reference was added.
Material and Methods
Figure 1: defined in the figure caption what is Q, and what is HRT
R = Thanks for the observation. Q and HRT were defined in Figure 1 caption
lines 130, 131: why were these minutes of UV radiation chosen? Why these H2O2 dosages? Please, clarify and add relevant information
R = Thanks for the observation. These minutes of UV radiation and H2O2 dosage were evaluated considering previous studies on recalcitrant compounds removal by AOP’s (mainly UV/H2O2). In such investigations, it is indicated that activation times of hydroxyl radicals occur between 10 and 180 min and the dosage of peroxide depends on the initial concentration of organic matter. Considering the concentration used in this research (22 mg/L TOC), it would correspond to about 65 mM, but in terms of optimization a higher and a lower concentration level were considered. In line with the relevant comment made by the reviewer, the text was amended to clarify the sentence (including references).
Figure 2: indicate in figure caption what is HP and SSHFCW
R= Thanks for the observation. HP and SSHFCW were defined in Figure 2 caption
Table 1: defined what is HRT
R = Thanks for the observation. The action was realized.
Line 169: indicated that it is synthetic cropWW
R = Thanks for the observation. “Synthetic” was added before “cropWW”
Line 170: why 60 days?
R = Thanks for the question. The authors agree that the sentence The hybrid system was operated for 60 days “suddenly appeared”
In fact, 60 days of operation time is considered a long time of experimentation for this type of system because they are laboratory scale, operated continuously and the synthetic water was prepared constantly. In addition, all analytical determinations had to be performed according to HRT (60 for HRT = 1d, 30 for HRT = 2d and 20 for HRT = 3d) and all this in triplicate.
Specifically, the systems evaluated were operated for 60 days to ensure that the systems achieved stabilization. In this regard, an important finding was the stabilization of the systems (virtually no fluctuations in organic matter removal) since before 40 days (Figure 8).
The authors consider that, despite the explanation provided, this is an important observation. In this respect, the sentence was improved to clarify the idea (including references).
Line 171: how was the methodology for malathion determination optimised. Optimisation parameters should be included or a reference should be added
R = Thanks for the observation. The authors think that there was a mistake, since the quantification of malathion was not optimized, but was performed following the analytical procedures of USEPA (2007). In other hand, operating parameters (such as temperature, injection volumes, purge time) were including as indicated by the reviewer.
Line 175: defined ORP
R = Thanks for the observation. ORP was defined as oxidation-reduction potential
Line 176: at what type samples were collected? only after the 60 days? the effluent of CW was discarded? There was always input of new AOP effluent to CWs?
R = Thanks for the observations. With the reviewer question the authors noticed that the information provided was not completely clear.
The samples collected were single grab samples, because this samples were collected and analyzed in the influent and effluent of each CW system (remembering that this influent is the effluent of the optimized advanced oxidation process). Since the system is operated under continuous flow, each time the time of interest elapsed, the sample was collected and/or analyzed. Thus, by the time the 60 days had elapsed, 60 determinations had been made for each measured parameter and in triplicate for the system operated with HRT = 1d, 30 (x3) for the system operated with HRT = 2d and 20 (x3) for the system operated with HRT = 1 d. With this information, Figure 8 was made. This procedure exceeds that recommended by the code of federal regulations (of USA) (Title 40, Chapter 1, § 133.102) stipulates that treatment plant performance must be evaluated by tabulating 30-day averages of plant effluent.
Regarding the waste generated (effluents generated). The country's legislation considers them to be hazardous and they were stored and collected by a specialized company for their treatment and final disposal. A small fraction was used for another research in which ultrasound treatment is being evaluated.
Results
Figurw4: this caption need more information. What is being shown in yy axis? what is Ce/Co? Removal in what units? Values presented are mean values with their standard deviation? how many replicates? all these information need to be in figure legend
R = Thanks for the observation. The authors apologize for not providing the information in the first version. In the new version all this information was included in the figure caption
Figure5 : this legend need more information, please, see comments for figure 4 and make the necessary amendments
R = Thanks for the observation. The authors apologize for not providing the information in the first version. In the new version all this information was included in the figure caption
Fiigure6: define pOR. Are these mean values?~
R = Thanks for the observation. “ORP” was included and defined instead of “pOR”. Also, mean values and number of replicates were added.
Table 2: how was this surface design analysis done? Please include relevant information
R = Table 2 only represents the most viable options from the results of the response surface design analysis (combined for malathion and TOC), since this design was made separately for each variable and is focused on the highest removal without considering the savings that could be made with a lower peroxide dose and shorter irradiation time. To clarify this idea, the title of the table was modified.
On the other hand, in Materials and methods section, before figure 2, the explanation on Response Surface methodology was slightly extended.
Line 263: indicate that it is synthetic CropWW
R = Thanks for the observation. Synthetic CropWW was indicated.
Line 275: “…by some mechanisms other than biodegradation” - this is not clear. Why not biodegradation? Please, clarify and add relevant information
R = Thanks for the observation. The authors had commented this statement without the pertinent clarification. The results in Figure 8 indicate that during the first 17 days of operation the CW behaved exactly the same behavior despite the difference between HRTs (1, 2 and 3 d). When biodegradation occurs, the removal efficiency is directly proportional to HRT as from day 17 where the difference is noticeably clear in Figure 8. In addition, it was the firt 17 days that bacteria within the CWs were in contact with the wastewater. The absence of biodegradation is a common behavior during the first days of operation. But an advantage or CW is that regardless of biodegradation, sorption phenomena exist as discussed in the text.
The authors hope that the explanation answers the observation. In the new manuscript, we added the reason why it is suggested that during the first 17 days biodegradation was not considered.
Figure 8: please, see comment for previous figures legend and include more information accordingly
R = Thanks for the observation. Information has been included in the figure legend, as was done with the previous figures. In addition, the meaning of each of the three lines was included.
Conclusions
Line314: indicate that it is synthetic croplant wastewater
Thanks for the observation. Indication was done.
Line 332: not only this but test should be done also with real wastewater. Please, include. Besides, some discussion should be done regarding the economical aspect in the sense that an hybrid system can be less costly than single AOP.
R = Thanks for the observation and suggestion. The point about of real wastewater was included in the new version of the document. Also, a discussion about the importance to carry out the economic comparison between the proposed system and one bases only on AOP, since the one proposed in this research is expected to be less costly.
Sincerely, thank you very much for all your observations made for the improvement of the manuscript.

Reviewer 3 Report
The main aim of this article is to assess the technical feasibility of a UV / H2O2-CW coupled system for high malathion and nutrient loads in wastewater.
The connection of the two systems, AOP and CW, was done simply by means of a buffer tank
The AOP means a UV / H2O2 treatment has been optimized separately.
The experiments were well designed, the vast majority of the results were compared, explained and interpreted with the published data of other authors.
The continuous operation of the assembly of the experimental system raises questions, since how could the 3 experimental CW systems connected in parallel be supplied from the mixing tank marked with a volume of 13 liters (Figure 1)? Their combined capacity: 11.37 + 5.68 + 3.79 = 20.84 L / h! The capacity of the reactor filling the mixing tank is 18.47 liters / day.
Some abbreviations / notation explanations are missing: pOR, ORP, Ce, Co.
143 „Therefore, the effective volume ratio was 11.37 L” - Volume ratio, but relative to what?
149 „One stem per m2 was planted as recommendations by Kadlec..” How could they implement when the surface of SSHFCW is: 0.55x0.20 = 0.11m2? The width of the SSHFCW is incorrectly reported as 2000 mm in the upper right portion of Figure 3.
Fig.5 How do you explain the increase in TOC? In particular, a TOC value higher than the initial ratio at the 49mM dose?
235 „Indeed, efficiencies of malathion and TOC removal with 65 mM were higher than with 82 mM H2O2, despite no statistical difference (P = 0.11). „ - This sentence needs further explanation.
245 „H2O2 dosage between 49 and 65 mM were the optimal conditions to achieve satisfactory removal efficiencies for the subsequent treatment stage” - the data referred to in the text cannot be identified by the marking in Figure 7.
Author Response
Reviewer 3
Comments and Suggestions for Authors
The main aim of this article is to assess the technical feasibility of a UV / H2O2-CW coupled system for high malathion and nutrient loads in wastewater.
The connection of the two systems, AOP and CW, was done simply by means of a buffer tank
The AOP means a UV / H2O2 treatment has been optimized separately.
The experiments were well designed, the vast majority of the results were compared, explained and interpreted with the published data of other authors.
The continuous operation of the assembly of the experimental system raises questions, since how could the 3 experimental CW systems connected in parallel be supplied from the mixing tank marked with a volume of 13 liters (Figure 1)? Their combined capacity: 11.37 + 5.68 + 3.79 = 20.84 L / h! The capacity of the reactor filling the mixing tank is 18.47 liters / day.
R = Thank you very much for the observation. Figure 1 had many editing errors. The main ones are that the volumetric flow rate of the CW was in L/h instead of L/d. Also, the volume of the mixing tank had a typographical error and should be 143 instead of 13 L (we did not write the 4). In fact, for these calculations a hydraulic balance was made in which the 60 days of operation of the hybrid system were estimated. We annexed this mass balance and a table where it specifies the hydraulic parameters of each stage of the system so that the information can be verified. In addition, the caption figure was modified to clarify the concepts.
https://1drv.ms/b/s!Amaan2EmcJZ_gu5o3UwTe4r07mokOA?e=nkVgfJ
Some abbreviations / notation explanations are missing: pOR, ORP, Ce, Co.
R = Thanks for the observation. In the new manuscript version, all abbreviations and notation are defined, especially in the figure captions.
143 „Therefore, the effective volume ratio was 11.37 L” - Volume ratio, but relative to what?
R = Thanks for the observation. The word "ratio" was out of place. During the writing process the authors "kept in mind” the word ratio because of the length and width description. We thank you for noticing this. In fact, in the third line of section 2.3. It's properly written. “ratio” was deleted in the text.
149 „One stem per m2 was planted as recommendations by Kadlec..” How could they implement when the surface of SSHFCW is: 0.55x0.20 = 0.11m2? The width of the SSHFCW is incorrectly reported as 2000 mm in the upper right portion of Figure 3.
R = Thanks for the observation. The authors this is a very embarrassing mistake in the manuscript. We are really lucky that the reviewer and the editor give us a chance to correct it. One stem was planted in the 0.11 m2!!! This information has already been corrected in the manuscript (before line149 and in Table 1, where the information was also defined/clarified in the title of the table. Again, the authors thank you for observing the error.
On the other hand, the value of the wetland width has been corrected in Figure 3.
Fig.5 How do you explain the increase in TOC? In particular, a TOC value higher than the initial ratio at the 49mM dose?
R= Thanks for the observation. It is a grat question. Considering that the quantification of TOC was done under the strict procedures of APHA (2009) and considering that the malathion molecule has an important content of sulfur (approximately 20%) and that at low UV irradiation times (less than 30 min and in which the increase of TOC is observed in Figure 5) the malathion probability is degraded to organic compounds such as thiocyanates that are one of the main interferences for the technique of TOC and TIC. This depends very much on the relationship between the kinetics of radical generation OH*, malathion degradation and degradation of each of the byproducts. In this respect, some links of articles are included here where similar behaviour to that of the present study is observed when the dose of peroxide was not adequate. Another interference well documented by the standardized methods of different countries or associations such as USEPA and APHA, is the presence of carbonates and bicarbonates that do occur when oxidizing organic carbon, but these were removed during the analytical phase as indicated by the methodology described.
https://doi.org/10.1016/0043-1354(96)00052-8
https://www.sciencedirect.com/science/article/abs/pii/S0043135498002954
https://www.ncbi.nlm.nih.gov/pmc/articles/PMC3758901/
235 „Indeed, efficiencies of malathion and TOC removal with 65 mM were higher than with 82 mM H2O2, despite no statistical difference (P = 0.11). „ - This sentence needs further explanation.
R = Thanks for the observation. The authors agree. A brief explanation, in the same sentence, was included about more efficiency observed with 65 mM H2O2 compared to 89 mM (with new reference).
245 „H2O2 dosage between 49 and 65 mM were the optimal conditions to achieve satisfactory removal efficiencies for the subsequent treatment stage” - the data referred to in the text cannot be identified by the marking in Figure 7.
R = Thanks for the observation. Figure 7 had an incomplete axis of ordinates. This figure was corrected/improved, as was the figure caption.
The authors are grateful for the reviewer's accurate comments and indications
